# Comparison of Depressive Symptoms between International and Domestic Students in a Japanese University: Pre- and Post-COVID-19 Pandemic

**DOI:** 10.3390/brainsci14050447

**Published:** 2024-04-29

**Authors:** Yuki Shiratori, Takafumi Ogawa, Miho Ota, Noriko Sodeyama, Tetsuaki Arai, Hirokazu Tachikawa

**Affiliations:** 1Department of Psychiatry, Faculty of Medicine, University of Tsukuba, 1-1-1 Tennodai, Tsukuba 305-8577, Ibaraki, Japan; 2University Health Center, University of Tsukuba Japan, 1-1-1 Tennodai, Tsukuba 305-8577, Ibaraki, Japan; 3Ibaraki Prefectural Medical Center of Psychiatry, 654 Asahi, Kasama 309-1717, Ibaraki, Japan; 4Department of Disaster and Community Psychiatry, Faculty of Medicine, University of Tsukuba, 1-1-1 Tennodai, Tsukuba 305-8577, Ibaraki, Japan

**Keywords:** COVID-19, depression, university students, international students, network analysis

## Abstract

Background: The COVID-19 pandemic led to significant lifestyle changes and uncertainties, triggering a secondary wave of mental health issues in society. University students are especially susceptible to mental disorders. International students are considered more vulnerable due to limited emotional and financial support from their families and difficulties accessing community support. Hence, we conducted a longitudinal analysis to compare depressive symptoms among international students before and during the pandemic. Methods: Data from depression screenings conducted at the University of Tsukuba in 2019 and 2020 were utilized. We included all students enrolled in 2019 who underwent health check-ups in both 2019 and 2020. Participants completed the Patient Health Questionnaire-9 (PHQ-9), enabling a comparison of item scores between domestic and international students. Psychopathological network analysis was employed to examine relationships among the items. Results: Prior to the pandemic, international students generally exhibited relatively good mental health compared with domestic students. During the 2020 pandemic, no significant difference was observed, but international students tended to demonstrate better mental health. However, network analysis revealed intergroup variations in the relationships among PHQ-9 items, with concentration problems and suicidal thoughts being more prominent among international students. Conclusion: This study’s findings suggest that young people studying abroad experience mental health crises similar to their domestic counterparts. Nevertheless, the patterns of these crises may differ between the two groups.

## 1. Introduction

The COVID-19 pandemic significantly affected global mental health, causing fear and anxiety as well as voluntary isolation as a preventive measure [1]. College students who are undergoing a challenging transition to adulthood are particularly vulnerable to mental health problems [2,3,4]. The transition from high school to university often leads to adaptation issues [5], and the combination of college enrolment and COVID-19 lockdown created intense stress for students. In our country, Japan, since the confirmation of the first COVID-19 case in January 2020, the number of infections has continued to increase. On 7 April 2020, the first state of emergency was declared, under which many universities were closed and in-person activities were replaced with online alternatives. Students refrained from going out and remained in their rooms. Subsequently, a total of four states of emergency were declared prior to September 2021.

Indeed, various countries reported an increase in university students experiencing anxiety and depression due to COVID-19 [6,7,8,9,10]. In 2022, UNESCO highlighted the significant impact of the pandemic on the education system, with university students being the most affected [11,12]. Students not only faced societal changes but also dealt with shifts in their learning environment, including increased online lectures and decreased face-to-face classes, leading to a decline in their overall quality of life [13].

The travel restrictions imposed during the pandemic further isolated international students, yet little research has focused on their situation in Japan [14,15,16]. Cross-sectional and longitudinal studies on the mental health of international students before and after the pandemic are limited. Ke et al. conducted a longitudinal investigation into anxiety and depression among Chinese international students at Australian universities [17]. With the onset of the COVID-19 pandemic, there was an increase in the prevalence of anxiety and severe depression. That study did not conduct a comparison with domestic students, leaving uncertainty regarding whether the effects were specific to international students or applicable to university students in general. In a study by Jamshaid et al., indicators of depression and anxiety worsened among Chinese international students after the pandemic, with youth and female gender identified as risk factors [18]. The baseline survey period labeled as “pre-pandemic” in this study spanned from December 2019 to April 2020, potentially including periods not strictly considered pre-pandemic, such as between the onset of the outbreak in China (December 2019) and the WHO’s recognition of the pandemic (11 March 2020). Razgulin et al. conducted a longitudinal study on Lithuanian international students, revealing anxiety in approximately one-third of the students and emphasizing the importance of social support [19]. However, their baseline data collection also occurred during the COVID-19 pandemic, precluding a comparison between pre- and post-pandemic periods.

In contrast, our study assessed detailed data on depressive symptoms in international students before and during the pandemic.

This study aimed to investigate the impact of the COVID-19 pandemic on depression among international students compared with domestic students in Japanese universities. Additionally, it sought to examine changes in the relationship of each depression indicator to gain insights into intervention points.

## 2. Methods

### 2.1. Procedure

At the Health Center of the University of Tsukuba, a mental health check-up is conducted for all students as part of the general health examination. The purpose of this check-up is to detect and prevent mental health problems in students. Since 2016, the center has been using a depression screening tool, the Patient Health Questionnaire-9 (PHQ-9) [20], as a health examination item. This center conducts primary screening by setting criteria (e.g., PHQ-9 total score > 9 points) and calling in the students who meet these criteria, followed by secondary screening consisting of a diagnostic interview with a psychiatrist. If a student needs professional treatment, the Health Center at the University of Tsukuba provides medical treatment in outpatient settings.

We utilized the results of the PHQ-9 screening conducted in April 2019 as pre-COVID-19 data, and the PHQ-9 results from 2020 as data from during the COVID-19 outbreak. In 2020, the health checkups that would usually have been conducted in April were postponed due to the need for COVID-19 infection control measures. Therefore, we conducted a survey in June using a cloud-based educational support service system for lectures, supplementing the annual PHQ-9 survey with questions about difficult situations under the influence of COVID-19. Specifically, in the 2020 survey, participants were asked to complete a questionnaire that included items on demographic factors such as gender and age, lifestyle factors such as smoking, alcohol consumption, and exercise, pre-COVID-19 going-out status, post-outbreak going-out status, economic status, presence of cohabitants, and the PHQ-9 questionnaire.

### 2.2. Items and Scales

The PHQ-9 employs a 4-point Likert scale using a range from 0 (not at all) to 3 (nearly every day), with total scores ranging from 0 to 27 points (9 items). A cut-off score of 10 points is recommended for evaluation [20]. A Japanese-language version has been standardized [21].

The survey asked about the frequency of going out using an eight-point scale for both pre- and post-COVID-19 periods in 2020, with higher numbers indicating greater isolation at home. The items were based on the questionnaire that the Japanese Cabinet Office administered in 2018 to investigate the quality of life of Japanese citizens. The frequency of alcohol consumption was assessed using a scale of 1 to 5 as follows: (1) never; (2) less than once a month; (3) 2–4 times a month; (4) 2–3 times a week; and (5) 4 or more times a week. Economic condition was asked on a scale of 1 to 5 as follows: (1) very bad; (2) bad; (3) ordinary; (4) good; and (5) very good.

### 2.3. Subjects

This study targeted students enrolled at the University of Tsukuba who underwent PHQ-9 assessment as part of their health check-ups for two consecutive years in 2019 and 2020. Specifically, participants comprised all students enrolled at the university in April 2019. Among them, those who underwent health check-ups in April 2019 were included, and exclusion criteria eliminated those who did not in 2020. Students with a different nationality to Japanese were identified as international, based on their nationality information registered in the university database. The results of the surveys in 2019 and in 2020 were aggregated by linking them to the ID information. In 2019, there were 17,697 enrolled students, of which 15,325 and 2372 were domestic and international, respectively. Among the 12,265 students who underwent health check-ups in 2019, 10,705 and 1560 were domestic and international, respectively. Among them, 6221 domestic students and 626 international students underwent health check-ups in 2020. This study obtained research consent through an opt-out method. This study was conducted with the approval of the University of Tsukuba Medical Ethics Committee (Approval No. 1567).

### 2.4. Statistical Methods

We compared scores of domestic and international students on each item and the total PHQ-9 using the Wilcoxon signed-rank and chi-square tests. We conducted a multiple regression analysis with the dependent variable being the PHQ-9 total score in 2020 during the COVID-19 pandemic and with the following independent variables: being an international student, demographic data, going-out behavior, and PHQ-9 in 2019.

The relationship between each item in the PHQ-9 was analyzed using psychopathological network analysis. Network analysis is a technique that has been used to visualize and analyze relationships between people or organizations based on graph theory, originally called social networks. In recent years, an approach called psychopathological network analysis has been developed, which analyzes the relationships between psychological variables without assuming latent variables. In this type of analysis, each item with a variable is treated as a node, statistically significant parameters associated with the variables are identified as edges, and the structure of the network is evaluated [22]. Thus, for both the international and domestic student groups, networks were formed for the PHQ-9 items, and centrality indices were calculated for each item to examine their importance within the network.

Statistical analyses were performed using IBM SPSS Statistics 26 for Windows. Network construction and centrality calculations were performed using “qgrqph” and “bootnet” packages [23].

## 3. Results

We conducted a comparison of demographic data between domestic and international students. International students tended to be older (*p*-value < 0.001), predominantly female (*p*-value < 0.001), and more likely to be graduate students (*p*-value < 0.001). Additionally, smoking was more prevalent among international students (*p*-value < 0.001), while regular exercise habits were more common among them (*p*-value = 0.001). On the other hand, international students reported lower frequencies of alcohol consumption (*p*-value < 0.001) and worse economic conditions (*p*-value < 0.001) compared with domestic students. Before the COVID-19 outbreak, international students had a higher frequency of going out compared with domestic students (*p*-value < 0.001), but this trend reversed after the outbreak (*p*-value < 0.001) (Table 1). Domestic students also exhibited a significant decrease in the frequency of going out post-outbreak compared with pre-outbreak (median (IQR): 6(5–6) vs. 1(1–2); Z, −65.653, *p*-value < 0.001), as did international students (6(4.76–6) vs. 1(1–2); Z, −19.498, *p*-value < 0.001).

We compared the PHQ-9 scores between domestic and international students for each year, pre-COVID-19 in 2019 and post-COVID-19 in 2020 (Table 2). In 2019, international students had significantly lower scores than domestic students for depressive mood, sleep disturbance, appetite change, suicidal ideation, and total PHQ-9 score. In 2020, international students had significantly lower scores than domestic students for depressed mood, feeling tired, and sleep difficulties, but significantly higher scores for anhedonia and psychomotor agitation/retardation. There was no significant difference in total scores.

In the multiple regression analysis, using the total score for PHQ-9 in 2020 as the dependent variable, being an international student did not have a significant effect (β = 0.01, *p*-value = 0.059) (Table 3). However, being older (β = −0.062, *p*-value < 0.001), habitual exercise (β = −1.072, *p*-value < 0.001), living with someone (β = −0.442, *p*-value < 0.001), and having a better economic status (β = −0.177, *p*-value = 0.006) had positive impacts on improving the PHQ-9 score in 2020. On the other hand, being female (β = 0.578, *p*-value < 0.001), having a lower frequency of going out before the pandemic (β = 0.16, *p*-value < 0.001), and having a higher PHQ-9 total score in 2019 (β = 0.556, *p*-value < 0.001) had a negative impact on the PHQ-9 score in 2020.

We then used network analysis to compare the PHQ-9 scores between domestic and international students for each year. Figure 1 shows the comparison of network centrality between 2019 and 2020 for international students. Strength centrality indicates how strongly a node is connected to other nodes in the network. Closeness centrality is the average distance from a node to all other nodes in the network. Betweenness centrality is the number of times a node is on the shortest path between two other nodes [23,24,25]. The top three items with the largest increase in terms of strength were concentration problems, appetite changes, and suicidal thoughts. The bottom three items in descending order were sleep difficulties, feelings of worthlessness, and depressed mood, which actually decreased for international students in 2020. For closeness, the top three items with the largest increase were psychomotor agitation/retardation, suicidal thoughts, and concentration problems. The bottom three items in descending order were feeling tired, feelings of worthlessness, and sleep difficulties. For betweenness, the top three items with the largest increase were suicidal thoughts, anhedonia, and sleep difficulties. The bottom three items in descending order were feelings of worthlessness, concentration problems, and psychomotor agitation/retardation, with decreases observed in 2020 for items 7 and 8.

Figure 2 shows a comparison of network centrality between 2019 and 2020 for the PHQ-9 of domestic students. For strength, the three items with the largest increase were feeling tired, anhedonia, and appetite changes. The items that increased least were feelings of worthlessness, suicidal thoughts, and psychomotor agitation/retardation; these three items decreased in 2020 compared with 2019. For closeness, the three items with the largest increase were feeling tired, concentration problems, and sleep difficulties. The three items with the lowest increase were psychomotor agitation/retardation, depressed mood, and suicidal thoughts. For betweenness, the three items with the largest increase were feelings of worthlessness, concentration problems, and feeling tired. The three items with the lowest increases were appetite changes, psychomotor agitation/retardation, and suicidal thoughts; these three items decreased from 2019 to 2020.

## 4. Discussion

In this study, the mental health of international students studying in Japan before and during the COVID-19 pandemic was compared. There have been no similar reports, to the best of the author’s knowledge. Mental health was also compared between international and domestic students at the same Japanese university.

In terms of the total scores on the PHQ-9, prior to the pandemic, the depressive symptoms of international students were relatively good compared with domestic students. During the 2020 pandemic, there was no significant difference, but international students tended to have better mental health. When 2019 and 2020 were compared, both international and domestic students showed worsening trends. In our previous report, the proportion of the entire student population showing symptoms of depression increased from 5% before to 10% during the pandemic [10]; here, we found that mental health among the international student subgroup also declined during the COVID-19 pandemic, which is consistent with the findings reported by Ke et al. [17].

The demographic data indicated that international students were older and a higher proportion of them exercised habitually. It has been reported that being younger [18,26,27] and not having exercise habits [28] were factors associated with worsened mental health during the COVID-19 pandemic. Thus, these attributes of international students appeared to serve as protective factors against worsened mental health during the COVID-19 pandemic. On the other hand, being female has somewhat inconsistently been reported as being associated with deteriorating mental health during the pandemic [26,27,29,30]. Moreover, alcohol and smoking habits are generally more commonly observed among university students who experience mental health issues [31]. The higher proportion of females among international students, as well as the presence of alcohol and smoking habits, were considered to contribute to the vulnerability of international students.

A more precarious economic status was reported to have a negative impact on mental health, both before and during the COVID-19 pandemic, as indicated by previous studies [31,32,33]. However, there are inconsistent findings, with some studies suggesting that high income may have also been a risk factor [15,29].

When comparing the degree of social isolation before and after the COVID-19 pandemic for domestic and international students separately, it was found that the degree of social isolation increased for both groups after the COVID-19 pandemic, and both groups voluntarily restricted their outdoor activities in line with government policies.

According to the results of multiple regression analysis, being an international student was not significantly associated with depression during the COVID-19 pandemic, and there was no significant difference between their mental-health-related response to this new situation and that of domestic students. It is possible that the low frequency of outdoor activities before the pandemic, along with the lower level of depression in the previous year, had a positive impact. In other words, pre-existing mental health may have influenced depression during the pandemic.

It is hypothesized that international students would have been more vulnerable to the impact of the COVID-19 pandemic due to difficulties in obtaining support from private social networks such as family, being prone to physical isolation, having smaller communities, and facing challenges in returning to their home countries depending on the situation there [3,34,35]. However, in this study, the mental health of international students during the pandemic did not significantly differ from that of domestic students, although it was adversely affected by the COVID-19 pandemic. This result may have been influenced by the fact that international students had relatively high levels of pre-existing mental health.

It has been suggested that international students may not seek mental health support as much and may not realize that support services such as counseling are available [36,37,38]; it is unknown whether this played a role in our results. On the other hand, international students are suggested to have higher levels of self-esteem, life satisfaction, and mental well-being, including lower rates of depression and anxiety, compared with domestic students [38,39,40]. Based on these findings, even though international students may have experienced difficulties in accessing support and feeling isolated due to the impact of COVID-19, their overall higher levels of baseline mental health may have offset some of these effects. Additionally, for students who did not originally seek help, the decrease in supply of assistance may have had a minimal impact. While this study did not measure help-seeking behavior, the lower levels of depression among international students prior to the pandemic suggest that their baseline mental health may have been relatively high.

The differences in the outbreak situation between the home countries of international students and Japan may have also had an impact. Of the 626 international students in this study, 382 (60%) were from China. As China was the originating point of the pandemic, the intensity of the emerging infection situation was greater there, and it is thought that isolation measures were stricter there than in Japan. This fact may have had a positive impact on the mental health of Chinese international students in Japan.

In each item of the PHQ-9, both domestic and international students showed a worsening trend in 2020 compared with 2019, but differences in the patterns were observed in the network analysis. For both international and domestic students, the centrality of concentration problems increased, but for suicidal thoughts, only international students showed an increase in centrality, which may reflect the situation of international students who were more likely to feel isolated due to the impact of COVID-19. Concentration problems, which were more central to the experience of international students, are directly related to students’ academic performance. Thus, it would be in the common interest of the schools and students to ensure that learning and language support are provided for international students in particular.

## 5. Limitations

The limitations of this study are as follows: The PHQ-9 serves as a screening scale for depression rather than a diagnostic tool. The distinction between domestic and international students was determined based on the nationality registered in the data used for this research. However, it is possible that there were students with foreign nationalities who were born and raised in Japan. Additionally, it is unclear whether the international students surveyed in 2020 were actually living in Japan during the research period. For example, some may have received remote instruction in their home country while others remained in Japan, unable to return home. Since we did not investigate residence status during the COVID-19 pandemic, there is a possibility of overestimation.

## 6. Conclusions

Generally, the act of studying abroad can be a significant challenge for young, inexperienced individuals, but it can also serve as a first step towards enhancing global understanding, international networks, and situational intelligence, ultimately uplifting the collective wisdom and prosperity of humanity. The outbreak of COVID-19 created barriers for studying abroad, but supporting this challenge is crucial. According to the results of this study, the mental health issues faced by international students were similar to those of domestic students, but the patterns differed somewhat. Considering the greater likelihood of isolation among international students, it is believed that they require more substantial support. The insights obtained from this study are deemed valuable for supporting the mental health of international students even in the post-COVID era.

This study’s findings suggest that young people studying abroad experience mental health crises similar to their domestic counterparts. Nevertheless, the patterns of these crises may differ between the two groups.

## Figures and Tables

**Figure 1 brainsci-14-00447-f001:**
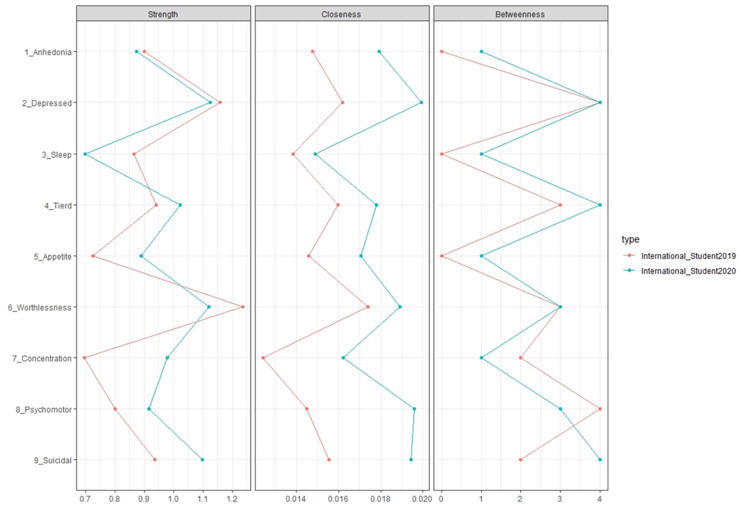
The PHQ-9 centrality scores between international students for each year. Legend: The changes in the importance of each item were visualized using network analysis. The red lines represent 2019, and the blue lines represent 2020. The higher the numerical value in the three centrality measures (moving towards the right), the greater the importance.

**Figure 2 brainsci-14-00447-f002:**
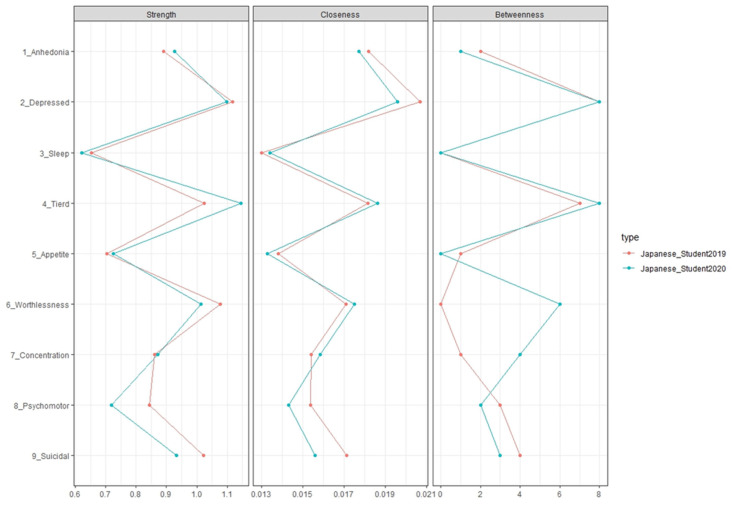
The PHQ-9 centrality scores between domestic students for each year. Legend: The changes in the importance of each item were visualized using network analysis. The red lines represent 2019, and the blue lines represent 2020. The higher the numerical value in the three centrality measures (moving towards the right), the greater the importance.

**Table 1 brainsci-14-00447-t001:** Comparison of demographics between domestic and international students.

Characteristic	International Students	Domestic Students	*p*-Value
	*n* = 626	*n* = 6221	
Age (mean ± S.D) ^a^	26.87 ± 4.3	22.23 ± 3.4	<0.001 **
female ^b^	324 (51.8%)	2574 (41.4%)	<0.001 **
Graduate student ^b^	478 (76.4%)	1305 (21.0%)	<0.001 **
No exercise habit ^b^	114 (18.2%)	1496 (24.0%)	0.001 *
Smoking ^b^	45 (7.2%)	216 (3.5%)	<0.001 **
Frequency of alcohol consumption (mean rank) ^c^	2917.56	3474.96	<0.001 **
Economic conditions (1: very bad–5: very good) (mean rank) ^c^	2846.49	3482.11	<0.001 **
Frequency of going out ^†^ before outbreak (mean rank) ^c^	3824.53	3383.7	<0.001 **
Frequency of going out ^†^ after outbreak (mean rank) ^c^	3253.81	3441.13	0.014 *

^a^: *t*-test; ^b^: χ2; ^c^: Mann–Whitney U test. *: *p*-value < 0.05; **: *p*-value < 0.01. ^†^: 1. Going out every day of the week for work or school. 2. Going out three to four days a week for work or school. 3. Going out frequently for leisure, etc. 4. Going out sometimes to socialize. 5. Usually staying at home and only going out on errands related to one’s hobbies. 6. Usually staying at home but going to the neighborhood convenience store. 7. Leaving one’s room but not leaving the house. 8. Hardly ever leaving one’s room.

**Table 2 brainsci-14-00447-t002:** The comparison of PHQ-9 between domestic and international students.

	2019				2020			
PHQ-9 Items	International Students	Domestic Students			International Students	Domestic Students		
	*n* = 626	*n* = 6221	*t*	*p*	*n* = 626	*n* = 6221	*t*	*p*
1. Anhedonia	0.31 ± 0.5	0.27 ± 0.5	−1.79	0.074	0.54 ± 0.7	0.46 ± 0.7	−3.034	0.002 **
2. Depressed mood	0.32 ± 0.5	0.4 ± 0.6	3.137	0.002 **	0.48 ± 0.6	0.57 ± 0.7	3.249	0.001 **
3. Sleep difficulties	0.48 ± 0.7	0.57 ± 0.8	3.13	0.002 **	0.7 ± 0.8	0.91 ± 0.9	6.454	<0.001 **
4. Feeling tired	0.62 ± 0.7	0.66 ± 0.7	1.198	0.231	0.68 ± 0.7	0.85 ± 0.8	5.337	<0.001 **
5. Appetite changes	0.3 ± 0.6	0.39 ± 0.7	3.771	<0.001 **	0.4 ± 0.6	0.44 ± 0.7	1.285	0.199
6. Feeling of worthlessness	0.22 ± 0.5	0.25 ± 0.6	1.161	0.246	0.36 ± 0.6	0.36 ± 0.7	−0.178	0.859
7. Concentration problems	0.22 ± 0.5	0.2 ± 0.5	−1.192	0.233	0.33 ± 0.6	0.3 ± 0.6	−1.078	0.281
8. Psychomotor agitation/retardation	0.15 ± 0.4	0.13 ± 0.4	−0.954	0.34	0.17 ± 0.5	0.11 ± 0.4	−3.009	0.003 **
9. Suicidal thoughts	0.04 ± 0.2	0.06 ± 0.3	2.339	0.02 *	0.07 ± 0.3	0.08 ± 0.3	0.676	0.499
10. Total	2.66 ± 3.0	2.91 ± 3.5	1.972	0.049 *	3.75 ± 3.9	4.08 ± 4.2	1.892	0.059

*: *p*-value < 0.05; **: *p*-value < 0.01.

**Table 3 brainsci-14-00447-t003:** Multiple regression analysis of the total score for PHQ-9 in 2020.

Variable	(Dummy Variable Code)	B	*t*	*p*	95% CI		VIF
Constant International student		4.944	11.176	<0.001	4.077	5.811	
	(Domestic: 0; international: 1)	0.01	0.059	0.953	−0.32	0.34	1.251
Age		−0.062	−4.005	<0.001	−0.093	−0.032	1.801
Sex	(Male: 0; female: 1)	0.578	6.443	<0.001	0.402	0.754	1.046
Exercise habit	(No: 0; more than 2–3 in a week: 1)	−1.072	−10.343	<0.001	−1.276	−0.869	1.029
Cohabitation	(Live alone: 0; with someone: 1)	−0.442	−4.814	<0.001	−0.622	−0.262	1.028
Smoking	(No: 0; smoking: 1)	0.128	0.551	0.582	−0.328	0.585	1.059
Frequency of alcohol consumption	(1: Never–5: four or more times a week)	−0.007	−0.174	0.862	−0.09	0.076	1.098
Graduate student	(graduate student: 1; others: 0)	−0.088	−0.66	0.509	−0.348	0.173	1.805
Economic conditions	(1: very bad–5: very good)	−0.177	−2.731	0.006	−0.305	−0.05	1.046
Frequency of going out before outbreak		0.16	4.419	<0.001	0.089	0.232	1.051
Frequency of going out after outbreak		−0.016	−0.63	0.529	−0.065	0.033	1.064
PHQ-9 total score in 2019		0.556	43.21	<0.001	0.53	0.581	1.039

Dependent variable: PHQ-9 total score in 2020. CI: confidence interval.

## Data Availability

The datasets used and/or analyzed during the current study are available from the corresponding author on reasonable request due to ethical reasons.

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
