# Peer review of "Comparison of Depressive Symptoms between International and Domestic Students in a Japanese University: Pre- and Post-COVID-19 Pandemic"

_brainsci, 2024, doi:10.3390/brainsci14050447_

Round 1

Reviewer 1 Report

Comments and Suggestions for Authors

The topic is important, but it has been well studied and presented in earlier publications of various research groups in the world. However, it does not detract from the contribution of the authors.

Unfortunately, this manuscript needs major revision, including technical correction.

Major comment:

What is the scientific novelty of your research?

It was held 4 years ago. Have you published these results before?

The tables and figures in this manuscript look like they were borrowed from previously published articles by the authors.

Technical comments:

There is no information about the authors.

There is no information about the affiliation of the authors.

There is no information about the corresponding author.

There is no abstract.

No keywords (at least five words).

Links to publications used by the authors should be in square brackets indicating the number in the order of citation.

The Introduction section is not informative, as it does not contain an explanation of the unresolved problems that prompted the authors to conduct this study.

The purpose of this study has not been formulated by the authors.

Subsection 2.3: There are no criteria for inclusion and exclusion of participants.

The presentation of the tables needs to be aligned with the MDPI template.

Low-quality figures need modification, improved pixel resolution and comments.

It is necessary to add the Limitations section after the Discussion section.

Comments on the Quality of English Language

A moderate correction of the data presentation style is desirable.

Reviewer 2 Report

Comments and Suggestions for Authors

The topic and design is acceptable, by and large. The manuscript needs several major revisions. My comments:

1. It`s a study on depressive symptoms, not mental health, amid COVID pandemic with pre-pandemic comparison not a longitudinal survey during COVID pandemic. In addtion, it`s a survey among all students not only international ones. All should be corrected in the title.

2. Data gathering and inclusion criteria should be described more in Abstract.

3. Introduction should be expanded, particularly previous studies and situation of Japan should be reviewed briefly.

4. What is '[1]' in line 32?

5. The manuscript needs exact proof-reading for instance line 43.

6. Is it correct: The PHQ-9 employs a 4-item Likert scale?

7. Citation should be corrected throughout the manuscript exactly.

8. Psychometric properties of  the Japanese version of PHQ-9 should be added with reference.

9. 'p=0' should be replaced with exact one.

10. First paragraph of Results should be extremely summarized.

11. Tables format should be consistent.

12. Fig-1&2 should be described clearly in legend.

13. 'In terms of the total scores on the PHQ-9, prior to the pandemic, the mental health of international students was relatively good compared to domestic students.' PHQ-9 shows depression, not mental health.

14. PHQ-9 is not a diagnostic tool. It should be pointed out as a limitation.

15. Conclusion should be a separate section.

Comments on the Quality of English Language

Minor editing of English language required

Round 2

Reviewer 1 Report

Comments and Suggestions for Authors

I thank the authors for responding to my comments.

The manuscript has been modified by the authors, but it still needs revision.

Major comments:

The results of this study have a low rate of novelty and relevance at the present time. The study was completed 4 years ago (data for 2019 and 2020). No catamnesis was presented by the authors in the following years (2021-2022). The authors published part of the material (on depressive disorders) earlier. I am very concerned about whether this article of this special issue will be cited in the future.

Minor comments:

Line 4 - Remove the quotation marks at the end of the title.

Lines 13-29: Please structure the abstract (background; purpose of the study; materials and methods; results; conclusion); in its present form, the abstract looks "vague", so this abstract will reduce the interest of potential readers of the journal in this article.

Lines 38-43: the fragment of the manuscript added by the authors needs correction of the font style; it is necessary to add links to this fragment.

Lines 52-69, 72-75: the fragment of the manuscript added by the authors needs correction of the font style.

Line 83: Add units of measurement (9 points).

Line 149 and further down the text: p-value.

Line 160: Add the name of the first column in Table 1 (for example, Characteristic or Parameter). Write all the characteristics in the first column of Table 1 with a capital letter.

Line 174: Add the name of the first column in Table 2 (for example, Mental disorders); remove the numbering of mental disorders (1, 2, 3, etc.).

Line 183: Modify the name of Table 3. What analysis have you presented here? Write the new name with a capital letter. All parameters in the first and second columns should be capitalized. Add a name to the second column.

The spelling and style of English needs revision. Also, pay attention to the spacing and dotting in your manuscript.

In summary, the manuscript needs to be seriously revised again.

I recommend that authors use the help of the English service of the MDPI publishing house.

Comments on the Quality of English Language

I recommend that authors use the help of the English service of the MDPI publishing house.

Reviewer 2 Report

Comments and Suggestions for Authors

No additional comment. Thank you!

Author Response

Dear reviewer 2,

Thank you for reviewing our manuscript. Your valuable comments have helped improve the quality of our paper. We appreciate it.

Sincerely,

Yuki Shiratori